# Prospects in the Use of *Cannabis sativa* Extracts in Nanoemulsions

**DOI:** 10.3390/biotech13040053

**Published:** 2024-12-02

**Authors:** Ian Vitola, Carlos Angulo, Raul C. Baptista-Rosas, Luis Miguel Anaya-Esparza, Zazil Yadel Escalante-García, Angélica Villarruel-López, Jorge Manuel Silva-Jara

**Affiliations:** 1Departamento de Ingeniería Química, Universidad de Guadalajara, CUCEI, Blvd. Marcelino García Barragán 1421, Olímpica, Guadalajara 44430, Jalisco, Mexico; ian.castro8806@alumnos.udg.mx (I.V.); zazil.esclante@academicos.udg.mx (Z.Y.E.-G.); 2Grupo de Inmunología y Vacunología, Centro de Investigaciones Biológicas del Noroeste, S.C. (CIBNOR), Instituto Politécnico Nacional 195, Playa Palo de Santa Rita Sur, La Paz 23096, Baja California Sur, Mexico; 3Departamento de Ciencias de la Salud-Enfermedad como Proceso Individual, CUTonalá, Universidad de Guadalajara, Nuevo Perif. Ote. 555, Ejido San José, Tateposco, Tonalá 45425, Jalisco, Mexico; raul.baptista@academicos.udg.mx; 4Hospital General de Occidente, Secretaría de Salud Jalisco, Av. Zoquipan 1050, Colonia Zoquipan, Zapopan 45170, Jalisco, Mexico; 5Centro de Estudios Para la Agricultura, la Alimentación y la Crisis Climática, Centro Universitario de los Altos, Universidad de Guadalajara, Rafael Casillas Aceves 1200, Tepatitlán de Morelos 47600, Jalisco, Mexico; luis.aesparza@academicos.udg.mx; 6Departamento de Farmacobiología, Universidad de Guadalajara, CUCEI, Blvd. Marcelino García Barragán 1421, Olímpica, Guadalajara 44430, Jalisco, Mexico; angelica.vlopez@academicos.udg.mx

**Keywords:** medicinal plants, in silico analysis, low-cost medicine, alternative uses, cannabidiol

## Abstract

*Cannabis sativa* plants have been widely investigated for their specific compounds with medicinal properties. These bioactive compounds exert preventive and curative effects on non-communicable and infectious diseases. However, *C. sativa* extracts have barely been investigated, although they constitute an affordable option to treat human diseases. Nonetheless, antioxidant, antimicrobial, and immunogenicity effects have been associated with *C. sativa* extracts. Furthermore, innovative extraction methods in combination with nanoformulations have been proposed to increase desirable compounds’ availability, distribution, and conservation, which can be aided by modern computational tools in a transdisciplinary approach. This review aims to describe available extraction and nanoformulation methods for *C. sativa,* as well as its known antioxidant, antimicrobial, and immunogenic activities. Critical points on the use of *C. sativa* extracts in nanoformulations are identified and some prospects are envisaged.

## 1. Introduction

*Cannabis* is a plant that belongs to the *Cannabaceae* family; it is annual and dioecious. Over time, since its appearance (possibly more than 5000 years ago), it has been used for purposes such as obtaining fibers and oils. Originally, *Cannabis sativa* is believed to be indigenous to Eastern Asia, particularly in Mongolia and southern Siberia [1]. Due to its cultivation for various uses, this plant has spread globally and is now found in many parts of the world, including America, Europe, and Asia. Its adaptability to different climates and soils has contributed to its cosmopolitan distribution. The plant thrives in diverse environments, which has led to significant morphological and chemical variations based on geographical location and cultivation practices. At the pharmacological level, compounds such as alkaloids, flavonoids, terpenoids, and cannabinoids stand out. Of the latter, more than 130 have been isolated in trichomes of *Cannabis sativa* [2]. The structures in trichomes can be divided into glandular and non-glandular. Within the non-glandular trichomes, two types are not linked to the production of terpenoids, whereas there are three types of trichomes in the glandular ones: (1) stalked, (2) sessile, and (3) bulbous [3,4]. They are made up of secretory cells and stems with different functions. In recent years, there has been a resurgence of interest in its therapeutic potential due to its complex chemical composition, including cannabinoids such as tetrahydrocannabinol (THC) and cannabidiol (CBD), among others [5]. Although several authors have proposed classifying them into three species, *C. sativa*, *Cannabis indica*, and *Cannabis ruderalis*, most research groups agree that it is a monotypic plant; that is, all varieties belong to the same species, *C. sativa*. Another classification is the chemotaxonomic one, which considers phenotypes based on the proportion of cannabinoids. However, even though this classification may not fully account for the cannabinoid content that different chemotypes can present, it is still considered a helpful tool for classification [6].

The *Cannabis sativa* plant, a subject of extensive research, holds immense promise in the modulation of the immune system [7], oxidative stress [8], and antimicrobial capacity [9]. These studies have revealed the potential of cannabinoids, terpenes, flavonoids, and lignans [10], which are metabolites present in the plant that can bolster the body’s antioxidant response, enhance antimicrobial treatments, and exert beneficial immunomodulatory effects [11,12,13]. Unlike isolated compounds, these phytochemicals present in extracts can also act synergistically to promote therapeutic effects, a phenomenon known as the entourage effect [14]. With approximately 500 compounds in *C. sativa*, the presence of THC in extracts has been a critical focus, leading to the study of cannabinoids such as CBD [15] and other terpenes such as β-caryophyllene (BCP). The potential benefits of these extracts are vast, and overcoming the challenges in their practical application, such as reducing the psychoactive effect of THC [16], preventing the degradation of these active compounds [17,18], and enhancing bioavailability [19], could revolutionize the field of *C. sativa* therapeutics.

The problem of the psychotropic stimulus caused by THC has been addressed by exploring other non-psychoactive cannabinoids, such as CBD, cannabigerol (CBG), cannabichromene (CBC), and cannabidivarin (CBDV) [20]. On the other hand, to reduce the degradation processes of active compounds, the development of protection methods such as nanoencapsulation has been promoted [21]. This approach protects cannabinoids, terpenes, and flavonoids from environmental effects (light, temperature, and oxygen) to prolong their stability and half-life [22]. Related to the above, the improvement of bioavailability has been studied to increase the therapeutic capacity of active compounds, highlighting that the oral route has the lowest assimilation rate [23,24]. For example, nanoemulsions are an alternative to the challenge of oral administration because they can protect bioactive compounds from digestive processes and improve their bioavailability and bioaccessibility.

Nanoemulsions have been manufactured from oils/lipids (olive oil, phospholipids, and sunflower oil) [25,26], proteins (albumins, soy protein isolate, whey protein isolate, β-lactoglobulin, rice protein isolate, and peanut protein) [27,28,29], polysaccharides (carboxymethylcellulose, chitosan, glucans, pectins, alginates, hyaluronic acid, starches, cellulose, xanthan, guar, and gum arabic) [30,31,32,33,34], surfactants (sodium lecithin, lactoglobulins, and sodium dodecyl sulfate) [35], and plasticizers (glycerol, sorbitol, propylene glycol, and polyethylene glycol) [36,37], which form nanostructures with improved solubility and absorption of bioactive compounds. This technology can protect bioactive compounds from *C. sativa* for diverse therapeutic purposes. In this context, this paper describes and discusses recent research on *C. sativa* extracts (ECs) and the potential of nanoemulsion formulation to protect bioactive compounds and improve their bioavailability and bioaccessibility.

## 2. Extraction Methods Reported for *C. sativa*

*C. sativa* is considered an excellent source of bioactive compounds (cannabinoids, terpenes, flavonoids, and lignans) that can be extracted through diverse techniques [38]. Extraction methods can be classified into three broad groups. Traditional methods are based on the use of solvents (such as Soxhlet and maceration), alternative methods [38], and conventional methods without the use of solvents [39]. The main extraction methods used for *Cannabis sativa* are depicted in Figure 1. Solvent-based extractions can be subclassified into those using alcohols (methanol and ethanol), hexane, and ethers [40,41]. In this type of extraction, the solid material crushed in the organic solvent is disposed of in different proportions, filtered, and stored. In extractions using ether, organic acids are used to adjust the pH [42], and continuous evaporation–condensation cycles can be used to improve efficiency [43] or dynamic maceration, which is more efficient [44]. In addition, there are other methods for making oil extractions in which the use of edible vegetable oils as a solvent is required, such as coconut [45], sunflower [46], and olive [47]. For example, *Cannabis* flowers are dried between 85 and 145 °C for 40 min in a convection oven, pulverized, coated with olive oil (1:10), stirred for 40 to 120 min, filtered, and stored [47]. Although widely used, this type of extraction is inefficient and, in some cases, can leave potentially toxic residues [48].

Alternative extraction methods differ from traditional ones mainly because of their higher efficiency. They use physical and chemical principles that optimize the process, allowing its use in the food, cosmetic, and research industries. There are ultrasound-assisted (UAE), microwave-assisted (MAE), pressurized liquid (PLE), supercritical fluid (SFE) [49], and *Cannabis* hydrodynamics extractions [50]. In the case of PLE and SFE extractions, CO_2_ and high pressure are used [51] to achieve better extractions [52]. For example, the conditions that have been recommended for cannabinoid extraction are T ≥ 31 °C, P ≥ 1015 psi for CO_2_, T ≥ 57 °C, P ≥ 2625 psi for CO_2_/Absolute Ethanol (95/5, *v*/*v*) at a flow rate of 3 mL/min and (a) an extraction time of 60 min for SFE and 4 mL/min and (b) 30 min of extraction for PLE [53]. On the other hand, ultrasound-assisted (UAE) and microwave-assisted (MAE) extraction techniques are similar to the previous ones in that they reduce the time and use of solvents while obtaining bioactive compounds [54]. UAE extraction uses acoustic cavitation to generate microbubbles that break up plant material, and MAE extraction uses rapid microwave heating to increase the solubility of compounds and facilitate the extraction of cannabinoids, terpenes, and other compounds of interest using sample–solvent ratios, times, and temperatures [55]. Finally, the hydrodynamic extraction of *Cannabis* requires freezing the plant material, converting it into a nanoemulsion using hot water, and using ultrasound to use the hydrodynamic force to break the cell wall; finally, the steps they follow are liquid–liquid extraction, centrifugation, and drying at low temperatures [56]. Despite having the name “Cannabis”, this method has been little used on this plant. For a generalized visual summary of extraction methods for *Cannabis sativa*, see Figure 2.

Some critical extraction points may be considered in the extraction methods for obtaining bioactive compounds or extracts from *C. sativa*, as discussed below.

The solvent is decisive in the extraction of bioactive compounds; the efficiency and quality of the product depend on it, and the compound to be used must be safe and environmentally friendly. Solvents such as hexane, butane, and ether have been widely used. However, they are highly flammable and toxic, representing potential health risks [48]. These solvents can leave traces in extracts, and synthesizing nanoemulsions with them can affect purity, safety, and quality [22]. Therefore, choosing a solvent that achieves a high extraction performance, is safe, and is environmentally friendly is a challenge. Supercritical solvents have been a convenient alternative; however, high costs can reduce their use widely [57]. On the other hand, vegetable oils and solvents offer the possibility of obtaining less expensive quality extracts that do not compromise human health or pose a danger to the environment. In addition, simultaneous use with other technologies, such as ultrasonic extraction, could improve performance [58].

The extraction process requires controlling variables like extraction time, temperature, pressure, the proportion of plant material and solvent, and stirring speed [59]. Carelessness in one of these variables could lead to the loss of sensitive compounds. In this sense, reproducing an optimal and quality process is difficult, representing a drawback. Automation strategies to optimize and control the critical variables of the extraction process could be a good target for exploration, thus constantly adjusting the conditions as required by the process with precision and efficiency and avoiding less loss of bioactive compounds, which translates into better extraction percentages and the quality of the phytochemical profiles of interest [60,61].

## 3. Nanoemulsions: Materials and Synthesis Methods

Nanoemulsions consist of the dispersion of two immiscible liquid phases, where one is dispersed in the other in the form of droplets on a nanometric scale (<100 nm). They are composed of an aqueous phase, an oil phase, and an emulsifier. Nanoemulsions are primarily prepared with an aqueous phase, an oily phase, and surfactants [62]. Polar materials frequently used in the aqueous phase include polyols [63], simple alcohols [64], proteins [65], and carbohydrates [66]. In the oil phase, acylglycerols [67], essential oils [68], vitamins [69], and others of a lipophilic nature [70] are used. Surfactants confer stability to nanoemulsions and include lecithin [71], lactoglobulin [72], sodium dodecyl sulfate [73], and saponin [37,74]. Although these materials have been shown to protect cannabinoids and other compounds of interest from *C. sativa*, the formulation must be optimized, and the release mechanisms must be considered.

Additionally, surfactants, or emulsifiers, play an essential role in stabilizing nanoemulsions. They hold the two immiscible phases, oily and aqueous, together. However, the fillers of bioactive compounds can interfere with the emulsifier or have a greater affinity for one of the phases, causing processes such as Ostwald maturation in the short term [54]. Concerning synthesis, each stage of the process deserves care, starting from the component’s homogenization to the fine emulsion’s final packaging. Factors such as temperature, mixing order, stirring time, and speed in the homogenization process influence droplet size and polydispersity [36]. In this way, a good experimental design is necessary, and the quality of the equipment must be guaranteed throughout the process.

The synthesis of nanoemulsions is not the last step of the process; stability is one of the most critical points, as the particle size, the zeta potential, the surfactant, and the composition of the dispersing phase condition it. Cremation, sedimentation, coalescence, and Ostwald maturation are consequences of a loss of stability and bioactive compounds [75]. If large nanoemulsion droplets lead to sedimentation/cremation, an unfavorable zeta potential leads to aggregation and a marked difference in droplet sizes. Aspects such as viscosity can affect how the emulsion can be applied or dosed in experimental animal models [54]. To state the above in physicochemical terms, at a biological level, nanoemulsions could present little assimilation if the components used are not related to the cells of the target tissue. The stability and viscosity of nanoemulsions should be explored with new surfactants that reduce the interfacial tensions of the phases without significantly increasing the nanoemulsion’s size or the continuous phase’s rheological properties. Sedimentation, on the other hand, the surface modification of the particles, could improve electrostatic repulsion and prevent the aggregation of droplets. However, it must be ensured that this functionalization does not alter the mechanism of action of the nanoemulsion [76]. Furthermore, the synthesis method significantly influenced nanoemulsion properties. In this context, high- and low-energy methods have been used to synthesize nanoemulsions (Figure 3), as described below.

### 3.1. High-Energy Methods Used for Nanoemulsions

High extraction methods require high-intensity forces and little use of surfactants to cause an increase in kinetic energy that breaks droplets in the oil phase and increases dispersion in the aqueous phase [78]. Commonly used high-energy methods are ultrasound, high-pressure homogenization, and microfluidization. Ultrasound uses high-frequency waves in the mixture, generating cavitation and the collapse of the microparticles, reducing the size to nanometer values. It is necessary to adjust the pulse’s time and intensity since the temperature increase can cause the degradation of the compounds of interest [54,79]. High-pressure homogenization passes the mixture through a small hole that breaks the droplets to nanometer-scale value sizes, so a precise flow rate and pressure control are required [54,80]. Finally, microfluidization makes the mixture transit at high speed through small channels to cause shear forces capable of breaking the droplets of the emulsions, turning them into nanoemulsions. Characteristics such as channel geometry and velocity determine conditions for achieving normal size distributions in nanoemulsions [54,81].

### 3.2. Low-Energy Methods Used for Nanoemulsions

Low-energy methods for nanoemulsions are recommended to avoid the degradation of thermolabile compounds. However, they may need other types of surfactants and a greater quantity to stabilize the nanoemulsions [54,78]. The most used are solvent evaporation, membrane emulsification, and spontaneous emulsification. In solvent evaporation, the organic solvent in the mixture, in which the oil phase and surfactants are located, evaporates in a controlled manner. Selecting the appropriate solvent is critical, and one with a low boiling point is recommended because it allows the use of low temperatures, such as ethanol, acetone, lactic-co-glycolic acid (PGLA) [82], and tetrahydrofuran [70]. Another critical consideration to review is solvent safety and compatibility. In membrane emulsification, the dispersion of the oil phase is achieved by forcing the mixture through a porous membrane with electrical or hydrodynamic forces. The main element of this method is the selection of the membrane with appropriate pore sizes, pressure values, flow rate, and temperature control to obtain stable and uniform nanoemulsions [83]. Spontaneous emulsification is achieved by correctly combining the components of the mixture so that, under specific conditions of pH, temperature, and concentration, stable and uniform nanoemulsions are formed without needing energy [84,85].

In general, the nanoemulsions achieved by these methods are stable and uniform, which is suitable for applications in the pharmaceutical industry. In addition, the combination of different techniques can be considered, which can improve stability and biodistribution. The possibility of preserving bioactive compounds and their properties drives continuous process refinement and optimization based on the size, distribution, stability, and controlled release of *C. sativa* emulsions.

## 4. Antioxidant Activity of *Cannabis sativa* Extracts and Nanoemulsions

The study of cannabinoids has become an area of great interest due to their therapeutic potential, in which antioxidant power stands out. Although *C. sativa* extracts (CSEs) contain several cannabinoids, little is known about their antioxidant properties acting together. No reports were found on the analysis of the antioxidant capacity of CSE, although some bioactive compounds reported in the plant were found. In this context, it must be noted that the antioxidant capacity of extracts is essential for therapeutic applications, but determining that capacity can be challenging due to the heterogeneity of bioactive compounds [86].

The antioxidant activity of CBD increases the concentration of anandamide that activates the peroxisome proliferator alpha (PPAR-α), which in turn participates in the regulation of the expression of the antioxidant enzyme superoxide dismutase [87]. In another study, CBD decreased reactive oxygen species (ROS), pro-inflammatory cytokines, and lipid peroxidation in LPS-stimulated C57BL/6J mouse microglial cells [88]. Similarly, CSE with 72% THC reduced ROS formation by 80% in differentiated SH-SY5Y neurons compared to purified CBD and THC [89]. Not only are cannabinoids found in CSE that may have antioxidant properties, but flavonoids, terpenes, and other phytochemicals known for their antioxidant properties are also present [90]. Because the bioavailability of cannabinoids varies according to the route of administration (i.e., 6% orally) [91], nanoemulsions could improve the stability and bioavailability of phytocannabinoids and other CSE compounds with antioxidant capacity; other studies have evaluated nanoemulsions of crude extracts or compounds with antioxidant capacity. In this regard, it was demonstrated that a nanoemulsion with a *Nigella sativa* seed extract containing thymoquinone had greater bioavailability and antioxidant capacity than pure extract. In the context of *C. sativa* crude extract, this has been evaluated for its antioxidant properties in nanoemulsions, the opposite of cannabinoids such as CBD, which has been widely studied [92,93,94,95,96]. Therefore, studying nanoemulsions with CSE to increase antioxidant capacity is an opportunity for future research; all the above is represented in Figure 4.

## 5. Effect of *Cannabis sativa* Extract and Nanosystems on Antimicrobial Activity

Among the therapeutic interests aroused by *C. sativa* are its antimicrobial properties. The improved distribution of nanoemulsions shows promise in encapsulated extracts in dose reduction for effective microbial activity, although few studies have been published. For example, extracts of different varieties of *C. sativa* had variable antimicrobial effects depending on their chemical composition and the concentration used on the microbial strains. The most significant inhibition of growth by *C. sativa* extracts was against *Bacillus thuringiensis, Staphylococcus aureus, Candida albicans, Micrococcus luteus, Pseudomosas protegens, Saccharomyces cerevisiae,* and *Fusarium eumartii* [92]. In another study, *C. sativa* seed extract inhibited the growth of pathogenic *Enterobacteriaceae* and the formation of *Staphylococcus aureus* biofilms without detecting an antimicrobial effect on probiotic bacteria of the genus *Bifidobacterium* and *Lactobacillus* [98].

*Cannabis* extract has been shown to have an antimicrobial effect on pathogenic strains, an ability that may be linked to cannabinoids and terpenes; however, the level of activity can vary according to various factors such as the concentration of the bioactive compounds, the sensitivity of the target strains [99], and the level of nanoemulsions, the ability of the structures to cross the membranes of the microorganisms, the rate of erosion of the nanoemulsion, and the action against the efflux pumps present in the bacteria [100]. Thus, it is necessary to determine the antimicrobial capacity in several contexts to know the antimicrobial activity level of the nanoemulsion. In this context, it is crucial to determine the antimicrobial capacity and the spectrum of action of nanoemulsion-based *C. sativa* extracts. Bioactive compounds can be specific to Gram-negative or Gram-positive bacteria, non-selective, and harmful to beneficial bacteria [98]. However, studies on specificity, concentration variation in nanoemulsions, and spectrum range are suggested by testing on various pathogenic and beneficial Gram-positive and Gram-negative strains. In this way, the mechanism of antimicrobial action can be better understood, allowing better applications in treating pathogenic strains or promoting beneficial ones.

Additionally, an innovative approach has been the biosynthesis of *C. sativa* nanoparticles with antimicrobial activities. Fungicidal effects were studied using solid lipid nanoparticles (SLNs) and chitosan against the fungus *Fusarium solani* with a lower concentration of pure extract [101]. The pure EC obtained consisted of 770.3 mg of THC, 72.1 mg of CBD, and 28 mg of CBN per gram of resin and various terpenes (α-pinene 9338 mg, β-pinene 3132 mg, β-myrcene 29,837 mg, p-cymene 4031 mg, ocimene 2.3 mg, linalool 2853 mg, BCP 7038 mg, and α-humulene 1660 mg per gram of resin) and chitosan SLNs interacted with the negative charges of cell membranes, facilitating release into cells. In this way, the combination of terpenes, cannabinoids, and chitosan exerted a fungicidal effect against the microorganism. The researchers synthesized four types of SLN: control (183 nm), chitosan (237 nm, SLNQ), *Cannabis* (146.9 nm, SLNC), and Cannabis–chitosan (152.4 nm, SLNCQ), with the latter having greater inhibition capacity (90%) at an extract concentration of 0.06 μg/mL. SLNCQs had an EC release rate of 42.4% over 24 h, higher than that found in SLNCs, with 26.4%. In another report, bimetallic gold and silver nanoparticles synthesized with *Vitis vinifera* canes and *C. sativa* residues had antimicrobial activity by reducing the growth and biofilm formation of *Pseudomonas aeruginosa* [102]. Although nanostructures are not strictly speaking nanoemulsions, they are nanosystems and allow them to be considered in exploring other antimicrobial applications. Figure 5 graphically represents all the above.

## 6. Effect of *Cannabis sativa* Extracts Nanoemulsions on Immunogenicity Activity

Another aspect of the therapeutic properties of *C. sativa* to consider is immunomodulation [103,104,105]; see Figure 6. One of the aspects to highlight is that pure extracts of *C. sativa* have possibly been little studied due to the presence of THC and its known psychotropic effect [106]. There are relatively few studies that include cannabinoid mixtures. Blanton and collaborators [107] reported that the CBD–BCP interaction modulated pain and inflammation in a murine model. In another study, indomethacin (IND) nanoemulsions were prepared with BCP (BCP–IND) in different proportions to test the anti-inflammatory effect in LPS-stimulated macrophages. The results of the trials showed that BCP–IND nanoemulsions reduced the production of pro-inflammatory cytokines more than only BCP or IND. This indicates that cannabinoids alone have therapeutic potential. Still, in conjunction with other cannabinoids or other drugs, they can be even more effective, suggesting the use of pure *C. sativa* extracts in O/W nanoemulsions. In research focused on the treatment of diabetes mellitus (DM), a condition characterized by high oxidative stress and alteration of the immune system [88], Wistar rats with *C. sativa* oil-induced DM (CSO) and nanoemulsions of *C. sativa* oil (NECSO) and metformin were used. The trials demonstrated that CSO and NECSO positively affected DM symptoms and reduced glucose in urine, blood, triglycerides, and low-density lipoproteins (LDL) [108]. A graphic summary of this section is presented in Figure 6.

Although research on *Cannabis* and its great potential to modulate the immune system has been growing, it still faces the difficulty of the wide diversity of bioactive compounds in the plant and its multiple interactions with cells [103,105,110]. Thus, the synergistic effect of around 500 compounds is not the same when isolated. Therefore, characterizing each of these compounds and understanding their therapeutic potential is challenging. Regarding immunomodulation, whether anti- or pro-inflammatory, one bioactive compound could have a favorable effect, while at the same time, another can act as an inverse agonist [111], so elucidating each compound’s effects requires rigorous and exhaustive approaches. Preclinical studies addressing in silico, in vitro, and in vivo assays using animal models and clinical studies in humans are a possibility to validate the immunomodulatory effects of extracts. The immune system’s high specialization makes it highly complex [112], so it is necessary to evaluate the interactions of these compounds with the different cell lineages and sublineages that make up the system, as well as the nanoemulsions based on *C. sativa* extracts’ immunomodulatory effects.

## 7. Computational Analysis of *Cannabis sativa* Compounds

Computational analyses, in the context of *Cannabis sativa* bioactive compounds (isolated or in nanoemulsions), are necessary to understand their therapeutic properties better. Knowledge of these metabolites at the molecular level allows a better understanding of the interactions of these compounds with receptor proteins, helping to elucidate their mechanisms of action; for the precision of the metabolites in the *Cannabis* plant related to its genes and metabolic pathways, databases are used as proposed by [113]. Computational research has been carried out on the compounds provided by *Cannabis*. Researchers performed a characterization to explain the quantitative structure–activity relationships (QSARS) between cannabigerol (CBG), especially in the 3D structure, and the biological activity, complementing it with the density functional theory (DFT) to understand the stability and activity of the cannabinoid. In this way, they concluded that the geranyl chains of CBG follow a tendency to wrap around the central phenolic ring, leaving the side chains forming hydrogen bonds with the para-substituted hydroxyl groups. This research was also extended to molecular dockings with cytochrome P450-3A4, observing the decreased inhibitory effect of CBG on the key enzyme in drug metabolism, allowing a better understanding of the activity and structure of the compound [114].

Regarding immunomodulation, a study of bioactive peptides in *Cannabis* seed hydrolysates (bioHPHS) was carried out to evaluate their anti-inflammatory capacity in silico with the results obtained from in vitro tests in human monocytes [115]. Once the peptides in two bioHPHS were identified, ten were selected and synthesized in silico to determine their immunomodulatory capacity in monocytes. The peptides DDNPRRF, SRRFHLA, RNIFKGF, VREPVFSF, QADIFNPR, and SAERGFLY showed very high immunomodulatory activity. Subsequently, other molecular dockings were carried out to determine the interaction of the TLR4/MD2 peptide, seeking to understand the action in the inflammatory cascade modulated by these proteins. In a similar investigation, the impact of the cannabinoids CBD, CBG, and cannabinol (CBN) on DNA methylation was analyzed to understand the regulation of gene expression and cellular function; in this case, the target proteins were the TET enzymes important in DNA demethylation processes. Through computational simulations, it was shown that cannabinoids would have the ability to chelate ferrous ions, which could potentially interfere with the enzymatic activity, notably in CBD and CBN.

In an antimicrobial context, the antimalarial activity of compounds present in *Cannabis* was addressed in silico to elucidate in vivo results of tetrahydrocannabivarin (THCV) because this cannabinoid had a high affinity with the α/β tubulin proteins of the pathogen *Plasmodium falciparum* [116]. Based on these results, the authors suggested that this cannabinoid (THCV) would inhibit microtubules, preventing the parasite from advancing through red blood cells. In this way, one can understand the importance of computational analyses in the promising study of bioactive compounds of *C. sativa,* thanks to the fact that they allow a better understanding of the therapeutic properties and, at the same time, offer a broader view of the molecular interactions between these metabolites and the target receptors. Figure 7 is a graphical representation of this.

Although computational analysis exhibited some advantages, some challenges cannot be solved due to computational analytics requiring high data processing capacity and intensive operations [117]. This requires state-of-the-art equipment that integrates graphics processing units (GPUs) and central processing units (CPUs) well, ensuring efficient analyses. However, resource-limited academic contexts can limit this equipment’s availability and cost. Once the appropriate hardware is configured, software with great analytical power is required; this includes data processing, visualization, and molecular modeling. These criteria constitute a challenge due to the variety, approach, learning curve, and costs of licenses [118]. In this context, artificial intelligence has gained space, and tools such as machine learning and data mining have proven to have great potential to detect patterns in data [119,120,121]; however, extensive knowledge of statistics, mathematics, and biological sciences is required. Because of these challenges, an interdisciplinary collaboration where mathematicians, physicists, chemists, engineers, and other specialties converge is necessary.

## 8. General Critical Points for *Cannabis sativa*-Based Nanosystems

Nanosystems have emerged as a technological proposal with wide application in biomedicine and food. These systems provide better active compound availability, distribution, and conservation [22]. However, *C. sativa*-based nanosystems seem underutilized in research, unlike purified cannabinoids, which have many applications and studies. This could impede the broad understanding of the therapeutic potential of the crude extract of the plant using nanoemulsions. The pharmaceutical industry could take part in the research now that discussions of legalizing the use of *Cannabis* are more frequent; this is convenient because it destigmatizes the use of the plant and supports the different applications of *Cannabis* in conjunction with biotechnology. In this way, the population could have more confidence in the therapeutic advantages and limitations of *Cannabis* nanoemulsions in various clinical contexts. The industry could participate in innovation in harnessing *Cannabis* and scaling up the production of nanoemulsions. Therefore, each new publication based on *Cannabis* nanoformulation should consider in vitro and in vivo release tests to extrapolate this technology, determining cytotoxicity, efficacy, and the best delivery method [122,123]. A key aspect is to evaluate the lipophilic nature of the extracts and convey them appropriately. From this point of view, lipid transporters in the formulations will be crucial for subsequent tests for both their release and toxicological properties [124]. In addition, regarding normative regulations, it is necessary to review the updates for human applications. New nanoformulations would be subject to scientific scrutiny and clinical trials approved by regulatory agencies. The regulatory framework differs from country to country, and the efforts of regulatory agencies such as the European Union and the FDA work in the face of the growing research based on cannabinoids and the current market demands.

The study of nanostructures, especially nanoemulsions, is highly anticipated. Although many of the techniques mentioned to produce nanoemulsions have not been applied, for the most part, in the study of EC, any advance in technology in terms of formulation, the combination of homogenization processes, or the improvement of interfacial properties could lead to much more stable and secure structures [125,126,127,128].

Finally, the possibilities of *C. sativa*-based nanosystems are vast, as cannabinoids are studied in neurosciences [129,130,131,132], cardiovascular health [133,134,135], gut motility [136,137,138,139], and inflammation [105].

## 9. Conclusions

*C. sativa* offers excellent therapeutic potential, including antimicrobial, immunomodulatory, and antioxidant properties. Aspects such as extraction and synthesis methods can determine the stability of nanostructures. However, despite all the benefits of the plant, studying it still presents challenges. Computational analyses make it easier to understand the properties of the compounds that comprise extracts and how they interact with molecular targets. Given all this, the pharmaceutical industry’s active participation is needed to advance and exploit other forms of distribution in animals and, finally, application in humans.

## Figures and Tables

**Figure 1 biotech-13-00053-f001:**
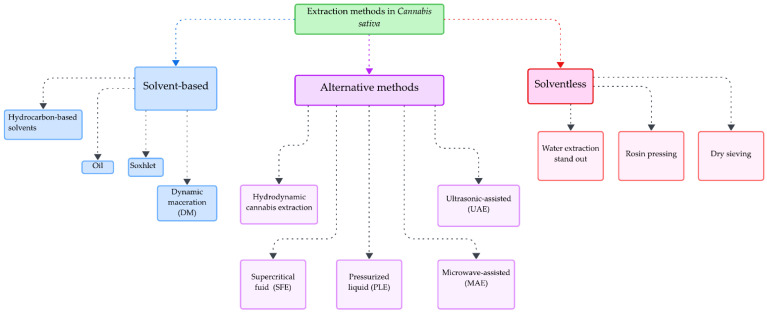
Extraction methods for *Cannabis*.

**Figure 2 biotech-13-00053-f002:**
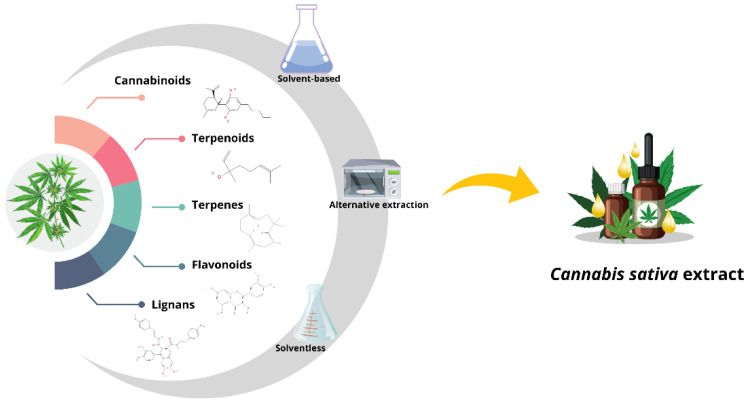
Methods that are traditional, alternative, and solvent-free. Each process can influence the efficiency and purity of the *Cannabis*’s bioactive compounds.

**Figure 3 biotech-13-00053-f003:**
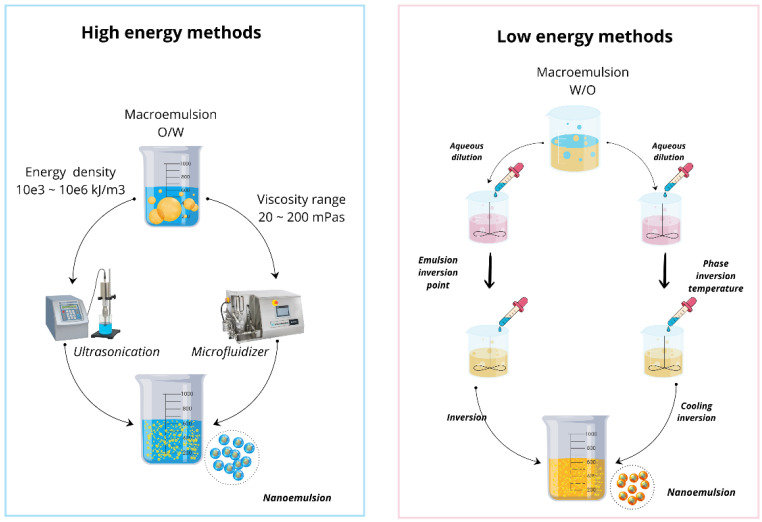
Nanoemulsion synthesis methods. High- and low-energy methods are the two groups into which they are classified. High-energy methods: the cavitation produced by the high frequency of the ultrasonic equipment breaks the emulsion droplets into smaller ones. Microfluidizers use high pressures that break up emulsion droplets. Low-energy methods: the emulsion inversion point (EIP) consists of changing the nature of the emulsion from water in oil (W/O) to oil in water (O/W), modifying the concentration and type of surfactants, varying the speed and inversion temperatures [54,77].

**Figure 4 biotech-13-00053-f004:**
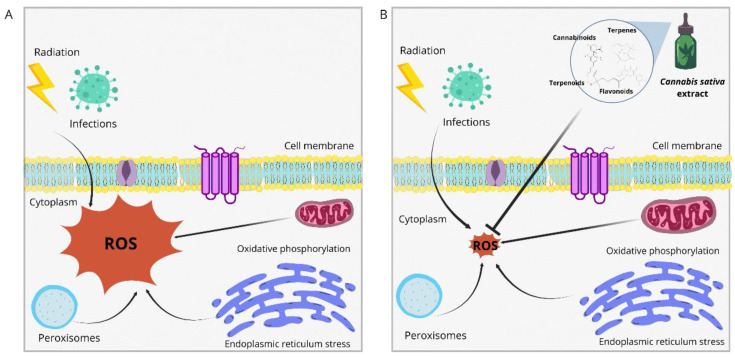
Antioxidant properties. ROS: reactive oxygen species. (**A**) Cell with a large amount of ROS produced by different factors. (**B**) The administration of EC is rich in antioxidant compounds capable of reducing reactive oxygen species [97].

**Figure 5 biotech-13-00053-f005:**
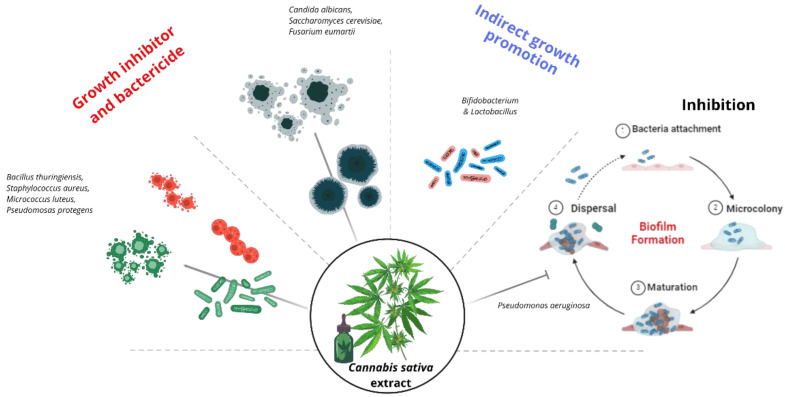
Antimicrobial activity. The EC inhibits bacterial growth, disrupting the bacterial membrane. It can also inhibit spore proliferation, but the effects at a structural level are unknown [98,100,101,102].

**Figure 6 biotech-13-00053-f006:**
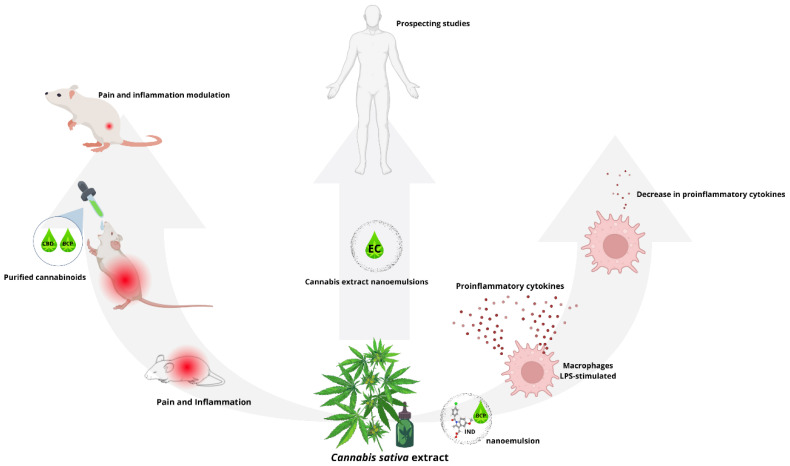
Immunomodulatory activity. The bioactive compounds cannabidiol (CBD) and caryophyllene (BCP) were administered orally in a murine model to treat pain and inflammation, reducing these conditions [108]. The mixture of indomethacin (IND) and BCP reduced the production of proinflammatory cytokines in a culture of macrophages stimulated by lipopolysaccharide (LPS) [109]. It is expected that studies with *Cannabis* extract nanoemulsions (NECs) will be carried out more frequently from now on.

**Figure 7 biotech-13-00053-f007:**
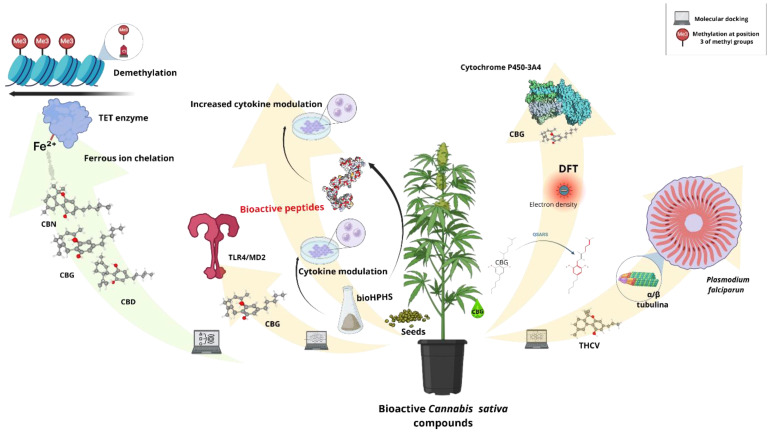
In silico studies. The analysis of computational studies on bioactive compounds from isolated metabolites and nanoemulsions of *Cannabis sativa* was parameterized in various functional modalities with comparative emphasis on therapeutic properties and molecular mechanisms. Such analyses include QSAR modeling, molecular docking, and DFT, which shed light on the degree of interaction between cannabinoids such as cannabigerol (CBG) with target proteins and their stabilities. One such example is CBG, whose inhibitory effects on cytochrome P450-3A4 were characterized [114]. Furthermore, some immunomodulatory peptides derived from *Cannabis* seed hydrolysates indicated some promising potential anti-inflammatory effects in silico and in vitro, especially since after the elucidation of the specific molecular mechanisms, they may prove useful as therapeutic agents [115]. Similarly, cannabinoids such as tetrahydrocannabivarin (THCV) were evaluated for their antimalarial activity by linking them to *Plasmodium falciparum* tubulin binding. These studies serve to underline the relevance of tools to explore the sources of bioactivity in *C. sativa* [116].

## Data Availability

No new data were created or analyzed in this study. Data sharing is not applicable to this article.

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
