# Peer review of "Prospects in the Use of Cannabis sativa Extracts in Nanoemulsions"

_biotech, 2024, doi:10.3390/biotech13040053_

Round 1

Reviewer 1 Report

Comments and Suggestions for Authors

The manuscript entitled 'Prospects in the use of Cannabis sativa Extracts in Nanoemulsions' is a critical survey that highlights the potential of the selected plant extracts with the integration of nanotechnology. However, the quality of the manuscript can be strengthened by addressing the following queries:

1)      In the abstract section's keywords, replace the abbreviation 'CBD' with its expanded form for better presentation.

2)      In the introduction section, consider discussing ethnobotany, taxonomy, and geographic locations of the selected plant.

3)      Regarding Figures 4 and 6, clarify whether they represent the proposed mechanism. If not, please cite relevant references.

4)      In Figure 7, rephrase 'In silico' to 'In Silico studies' for better clarity. Throughout the manuscript, use full terms instead of abbreviations for enhanced presentation. 5)      In Section 7, 'Computational analysis of Cannabis sativa compounds,' provide database details where relevant library information can be retrieved. To broaden the survey's scope, discuss the following studies: Cai S, Zhang Z, Huang S, Bai X, Huang Z, Zhang YJ, Huang L, Tang W, Haughn G, You S, Liu Y. CannabisGDB: a comprehensive genomic database for Cannabis Sativa L. Plant Biotechnol J. 2021 May;19(5):857-859. doi: 10.1111/pbi.13548. Epub 2021 Feb 4. PMID: 33462958; PMCID: PMC8131054.

6)      Additionally, complete Reference 44 and ensure adherence to author guidelines for uniformity.

Author Response

Responses to reviewers

Reviewer 1

The manuscript entitled 'Prospects in the use of Cannabis sativa Extracts in Nanoemulsions' is a critical survey that highlights the potential of the selected plant extracts with the integration of nanotechnology. However, the quality of the manuscript can be strengthened by addressing the following queries:

Thank you so much for your comments.

Comments 1:

In the abstract section's keywords, replace the abbreviation 'CBD' with its expanded form for better presentation.

Response 1:

Done.

Comments 2:

In the introduction section, consider discussing ethnobotany, taxonomy, and geographic locations of the selected plant.

Response 2:

Done. It was added to the new version attached.

Comments 3:

Regarding Figures 4 and 6, clarify whether they represent the proposed mechanism. If not, please cite relevant references.

Response 3:

Yes, we created the figures representing the mechanism described in the text and added their references.

Comments 4:

In Figure 7, rephrase 'In silico' to 'In Silico studies' for better clarity. Throughout the manuscript, use full terms instead of abbreviations for enhanced presentation.

Response 4:

Done.

Comments 5:

In Section 7, 'Computational analysis of Cannabis sativa compounds,' provide database details where relevant library information can be retrieved. To broaden the survey's scope, discuss the following studies: Cai S, Zhang Z, Huang S, Bai X, Huang Z, Zhang YJ, Huang L, Tang W, Haughn G, You S, Liu Y. CannabisGDB: a comprehensive genomic database for Cannabis Sativa L. Plant Biotechnol J. 2021 May;19(5):857-859. doi: 10.1111/pbi.13548. Epub 2021 Feb 4. PMID: 33462958; PMCID: PMC8131054.

Response 5:

Done. An explanation and reference were added to the new version in section 7.

Comments 6:

Additionally, complete Reference 44 and ensure adherence to author guidelines for uniformity.

Response 6:

Done. It was added as follows according to the author's guidelines:

Discover Advanced Hydrodynamic Cannabis Extraction - Cannabis Tech 2018. Available at: https://cannabistech.com/articles/hydrodynamic-cannabis-extraction/ (Accessed: 24 November 2024).

Reviewer 2 Report

Comments and Suggestions for Authors

very interesting work, no comments to add.

Author Response

Responses to reviewers

Reviewer 2

very interesting work, no comments to add.

Thank you so much for your time.

Reviewer 3 Report

Comments and Suggestions for Authors

The review article is well-written and demonstrates a high standard of scholarly writing. The content is well-researched, effectively addressing the aspects of nano-formulation methods for C. sativa. The review article is suitable for publication without requiring any revisions.

Author Response

Responses to reviewers

Reviewer 3

The review article is well-written and demonstrates a high standard of scholarly writing. The content is well-researched, effectively addressing the aspects of nano-formulation methods for C. sativa. The review article is suitable for publication without requiring any revisions.

Thank you so much for your time.

Reviewer 4 Report

Comments and Suggestions for Authors

Interesting article concerning extraction and delivery of active cannabinoids using nanoemulsions. Authors provided data about potential influence of cannabinoids functionalised nanoemulsion on different levels: antioxidant, antibacterial, immunomodulatory. All data are presented in interesting way.

Are there any regulations that will influence introduction of loaded nanoemulsions in practice?

What about potential cytotoxicity of cannabinoid nanoemulsions? What about potential delivery methods?

Author Response

Responses to reviewers

Reviewer 4

Interesting article concerning extraction and delivery of active cannabinoids using nanoemulsions. Authors provided data about potential influence of cannabinoids functionalised nanoemulsion on different levels: antioxidant, antibacterial, immunomodulatory. All data are presented in interesting way.

Thank you so much for your comments and time.

Comments 1:

Are there any regulations that will influence introduction of loaded nanoemulsions in practice?

Response 1:

The following information was incorporated into the Ms (section 8. General critical points for Cannabis sativa-based nanosystems).

“New nanoformulations would be subject to scientific scrutiny and clinical trials approved by regulatory agencies. The regulatory framework differs from country to country, and the efforts of regulatory agencies such as the European Union and the FDA work in the face of the growing research based on cannabinoids and the current market demands.”

Comments 2:

What about potential cytotoxicity of cannabinoid nanoemulsions? What about potential delivery methods?

Response 2:

The following information was incorporated into the Ms (section 8. General critical points for Cannabis sativa-based nanosystems).

Therefore, each new Cannabis nanoformulation should consider in vitro and in vivo release tests to extrapolate this technology, determining cytotoxicity, efficacy, and the best delivery method (Mobaleghol et al., 2024; Hernán Pérez de la Ossa et al., 2013). A key aspect is to evaluate the lipophilic nature of the extracts and convey them appropriately. From this point of view, the lipid transporters in the formulations will be crucial for subsequent tests for both their release and toxicological properties (Reddy  et al., 2023).